# Prodiginines Postpone the Onset of Sporulation in *Streptomyces coelicolor*

**DOI:** 10.3390/antibiotics9120847

**Published:** 2020-11-26

**Authors:** Elodie Tenconi, Matthew Traxler, Déborah Tellatin, Gilles P. van Wezel, Sébastien Rigali

**Affiliations:** 1InBioS—Centre for Protein Engineering, Institut de Chimie B6a, University of Liège, B-4000 Liège, Belgium; elodie@hedera22.com (E.T.); Deborah.Tellatin@uliege.be (D.T.); 2Hedera-22, Boulevard du rectorat 27b, B-4000 Liège, Belgium; 3Department of Plant and Microbial Biology, University of California, Berkeley, CA 94720, USA; mtrax@berkeley.edu; 4Molecular Biotechnology, Institute of Biology Leiden, Leiden University, 2333 BE Leiden, The Netherlands; g.wezel@biology.leidenuniv.nl

**Keywords:** programmed cell death, bacterial development, role of antibiotics, cell differentiation, morphogenesis, streptorubin, undecylprodigiosin

## Abstract

Bioactive natural products are typically secreted by the producer strain. Besides that, this allows the targeting of competitors, also filling a protective role, reducing the chance of self-killing. Surprisingly, DNA-degrading and membrane damaging prodiginines (PdGs) are only produced intracellularly, and are required for the onset of the second round of programmed cell death (PCD) in *Streptomyces coelicolor*. In this work, we investigated the influence of PdGs on the timing of the morphological differentiation of *S. coelicolor*. The deletion of the transcriptional activator gene *redD* that activates the *red* cluster for PdGs or nutrient-mediated reduction of PdG synthesis both resulted in the precocious appearance of mature spore chains. Transcriptional analysis revealed an accelerated expression of key developmental genes in the *redD* null mutant, including *bldN* for the developmental σ factor BldN which is essential for aerial mycelium formation. In contrast, PdG overproduction due to the enhanced copy number of *redD* resulted in a delay or block in sporulation. In addition, confocal fluorescence microscopy revealed that the earliest aerial hyphae do not produce PdGs. This suggests that filaments that eventually differentiate into spore chains and are hence required for survival of the colony, are excluded from the second round of PCD induced by PdGs. We propose that one of the roles of PdGs would be to delay the entrance of *S. coelicolor* into the dormancy state (sporulation) by inducing the leakage of the intracellular content of dying filaments thereby providing nutrients for the survivors.

## 1. Introduction

Microorganisms are offering us many natural compounds used for their therapeutic properties including antibacterial, antifungal, and antitumor agents, amongst many others. The most prolific microbial source of these natural products is the filamentous bacteria that belong to the genus *Streptomyces* [1,2]. Streptomycetes are Actinobacteria that have a complex multicellular lifestyle [3,4,5,6]. They grow as a branched mycelium consisting of vegetative or substrate hyphae. When nutrients become scarce or upon sensing of physico-chemical stresses, the bacteria enter into a differentiation process. This morphological change starts with the formation of an aerial mycelium emerging from the substrate mycelium and ends with the differentiation of the aerial hyphae into unigenomic spores that have a reproductive and disseminating function. When reproduction is required, an aerial mycelium is formed that feeds on the substrate mycelium, and eventually the aerial hyphae differentiate into long chains of spores. The production of secondary metabolites such as antibiotics is temporally correlated to the onset of morphological differentiation [7,8,9,10]. Why *Streptomyces* produce so many of these bioactive small molecules, and what the functions of these compounds for their producer, has only recently gained attention [11,12,13,14,15,16,17,18,19,20,21]. Two main conflicting viewpoints have stirred debate within this field, with some hypothesizing that ‘antibiotics’ are only weapons involved in territorial warfare between (micro)organisms, and others suggesting that, at the concentration that they are likely to be found in their natural environments, antibiotics might act as signalling molecules involved in intra- or interspecies communication [11,12,14,15,16,17,18,20,22]. Of course, these possibilities are not mutually exclusive, as one could imagine multiple context-specific functions for a single molecule. In addition, more complex roles are also attributed to antibiotics including to induce membrane damage of the producer to fulfil nutrient and energy needs, especially phosphate sources, for the surviving population [20,23].

Understanding the ecological functions of the ‘parvome’, i.e., the small molecules produced by microorganisms [24,25], remains a challenging task given the range of biological activities reported for these molecules. Most *Streptomyces* live in environments that are also home to many other bacteria, fungi, and plants [22], and so perhaps it is not surprising that they produce compounds with antibacterial, antifungal, and herbicidal properties. Finding a rationale for certain *Streptomyces* and other bacterial species to produce molecules that display anti-proliferative activities against cancer cell lines is much more puzzling. Recently, it was shown that anticancer compounds may also form a line of defense against phage infection [26]. We postulate that these specialized metabolites are also involved in controlling cell proliferation for the proper development (growth and differentiation) of the producing organism.

*Streptomyces* and other Actinobacteria produce DNA-damaging antitumor antibiotics such as molecules anthracyclines, bleomycins, enediynes, mitomycins, and prodiginines [27]. These molecules are so toxic that several mechanisms of self-resistance are often required for the survival of the producer, including toxin sequestration, efflux, modification, inactivation, self-sacrifice, and target repair [27]. Amongst this type of compounds, tri-pyrrole prodiginines (PdGs) are also able to disrupt the plasma membrane via a chaotropicity-mediated mode-of-action in addition to DNA cleavage [28,29,30]. Streptorubin B and undecylprodigiosin produced by *S*. *coelicolor* [31] are particularly interesting because these molecules remain trapped into the cytoplasm and the membranes of the bacterial filaments and are therefore likely to be necessary for the physiology of the producer rather than being involved in warfare with neighboring microorganisms. Moreover, in contrast to other DNA-damaging antitumor antibiotics [27], the biosynthetic gene cluster (BGC) of PdGs does not include genes involved in self-resistance. Why does *S*. *coelicolor* internally produce highly toxic compounds without having genes for self-resistance? These two exceptional features, i.e., (i) no secretion and (ii) no resistance genes, prompted us to investigate a possible role in the programmed cell death (PCD) process of the producer. PCD is a hallmark of multicellularity in bacteria, such as in *Myxococcus*, cyanobacteria and Actinobacteria, as well as in biofilms [6,13,32,33,34,35,36,37]. A study of the cyanobacterium *Microcystis aeruginosa* showed that the external addition of prodigiosin induced the formation of reactive oxygen species, which induced PCD [38,39]. Consistent with a role in the onset of PCD, the production of PdGs in *S*. *coelicolor* perfectly follows in time and space the zone of the culture where filaments were submitted to a first round of PCD [13]. Once produced, PdGs themselves inflict a massive second round of death resulting in an important destruction of essential macromolecules (DNA, RNA, and proteins) [13]. In contrast, a PdG non-producing mutant (strain *S*. *coelicolor* Δ*redD*) accumulates viable biomass and macromolecules confirming the occurrence of a PdG-dependent destructive process of the vegetative mycelium in the wild-type strain at this stage of the life cycle [13]. 

What is the role of PdGs in the morphological differentiation of *S*. *coelicolor* and what is the fate of the mycelium that survived the round of PCD associated with PdG production? In this work we attempted to provide the first evidence to solve these questions by assessing the consequences of deletion, or instead, the addition of multiple copies of the *redD* transcription activator on the morphological differentiation of *S*. *coelicolor*, and by monitoring PdG production in PCD survivors.

## 2. Results

### 2.1. Inhibition of PdGs Synthesis Causes Precocious Onset of S. coelicolor Morphological Differentiation

The life cycle of *Streptomyces* includes radical morphological and physiological changes, particularly at the transition between the vegetative (growth) and the aerial (reproductive) life styles. The onset of PdG production coincides with the transition phase (~40 h) [13] and, in order to assess the impact of PdGs on the development of *S. coelicolor*, we compared the timing of aerial hyphae formation between the wild-type strain *S. coelicolor* M145 and its *redD* mutant M510; the latter is unable to produce PdGs. As shown in Figure 1a, the surface of the culture of the *redD* mutant grown on the R2YE medium presented the first white spots corresponding to aerial hyphae formation after only 40 h of growth. The surface of the plate was entirely covered with aerial hyphae between 4 and 10 h later.

The precocious erection of aerial hyphae in the *redD* mutant was confirmed by scanning electron microscopy (SEM) (Figure 1b). Indeed, after 48 h, the parental strain (WT) still primarily produced vegetative hyphae, with only very rare evidences of erect aerial hyphae (Figure 1b, panels a and b), at a time when the *redD* mutant had already formed abundant aerial hyphae (Figure 1b panel c), with occasional spore chains (Figure 1b panel d). At 72 h, the wild-type strain produced erect aerial hyphae (Figure 1b panel e), while the aerial hyphae of the *redD* mutant had fully differentiated into mature spore chains (Figure 1b panel f).

In contrast, when strain *S*. *coelicolor* M510 was complemented with the low-copy-number plasmid pELT003 containing *redD* with its upstream region [13] (strain ET003, see Methods section), we observed both overproduction of PdGs (due to the higher number of copies of *redD*), and an important delay of the sporulation process (Figure 2). Indeed, strain ET003 presents a phenotype very similar to the bald mutants of *S*. *coelicolor*, which are stalled early during development and are unable to erect aerial hyphae.

Finally, as additional evidence that PdGs postpone sporulation of *S. coelicolor* we applied phosphorylated sugars to the R2YE medium, which are known to delay the occurrence of the second round of PCD in *S*. *coelicolor* and thus PdG production. As shown in Figure 3, the supply of glucose-6-phosphate (Glc-6P) or glucosamine-6-phosphate (GlcN-6P) to R2YE medium reduces the production of PdGs which ultimately results in early erection of aerial hyphae.

The early triggering of the developmental program of strain M510 is also accompanied by an early activation of secondary metabolism, in line with the temporal connection between the two processes. Indeed, actinorhodin production was accelerated by 12 h in the PdG-non-producer M510 as compared to the parent M145, resulting in higher accumulation of this blue-pigmented antibiotic (Figure 4).

### 2.2. Inhibition of Prodiginine Synthesis Accelerates Transcription of the Late Developmental Genes

Given the accelerated development of aerial hyphae in the *redD* mutant, we wondered whether this acceleration was due to the enhanced expression of key developmental genes. To test this possibility, *S. coelicolor* M145 and its *redD* mutant were grown as dispersed colonies and allowed to initiate the developmental program. At 68 h, the parental strain and its *redD* mutant showed clear morphological differences; the *redD* mutant colonies were almost completely covered with a thick layer of aerial hyphae, while the wild-type colonies had only begun to produce aerial hyphae (top panels in Figure 5). At this time point, we harvested and lysed the colonies, and quantified total RNAs in the cell lysates using the Nanostring nCounter analysis system as described previously [19]. We tested the expression of 11 genes that span different phases of the developmental program (see heatmap in Figure 5). We considered genes to be differentially expressed if they were at least two-fold different in their expression level, and had *p* values ≤ 0.02 in a Student’s *t*-test.

Multiple developmental genes showed differential expression in the *redD* mutant strain. Four developmental genes showed enhanced expression in the *redD* mutant: *bldN*, *chpA*, *rdlA*, and *whiE*. BldN is a key sigma factor that is active at a midpoint in the developmental program, and it is required for the raising of aerial hyphae [40]. ChpA and RdlA are representatives of the chaplins and rodlins, respectively, which are small proteins that form a hydrophobic coat on the surface of aerial hyphae and spores [41,42,43,44]. Expression of *chpA* and *rdlA* is dependent upon BldN [45]. Chaplins, together with SapB, are also biosurfactant peptides that reduce the surface tension at the colony-air interface in order to allow the erection of aerial filaments from the vegetative mycelium [43]. SapB is the lantibiotic-like peptide resulting from the post-translational modification by RamC from the product of the gene *ramS* [46]. SapB production is the very last step of the developmental cascade in *S*. *coelicolor* and is absolutely required for aerial formation in rich media [43]. However, *ramC* and *ramS* have contrasting expression patterns in the *redD* null mutant (Figure 5). This prevented us from forming a conclusion on the overall expression of the *ram* gene cluster and therefore on the production of SapB. We therefore could not predict whether the accelerated formation of aerial hyphae in the *redD* mutant would result from the overproduction of either both chaplins and SapB or only chaplins. Finally, enhanced expression of the *whiE* cluster for the type II polyketide grey pigment of the spores was also indicative of the early expression of a late sporulation checkpoint [47]. 

Interestingly, the transcription of *bldA*, which specifies a leucyl tRNA required for translating the rare UUA codon that is present in many developmental genes [48] was reduced, whereas that of *bldH* (*adpA*), which encodes a key positive regulator for multiple developmental genes [49], was unaffected in the *redD* mutant strain. These two genes are among the first genes of the developmental cascade and are expressed much earlier than SapB and chaplin encoding genes. Since at 68 h, the *redD* mutant was fully covered with aerial hyphae, and thus SapB and chaplins production already occurred, we hypothesize that the early part of the cascade involving *bldA* and *bldH* had already occurred and that their expression had already peaked and receded.

Beyond developmental genes, we also observed an increased expression of the actinorhodin biosynthetic gene *actI-*ORF1 in the *redD* mutant. Taken together, the enhanced expression of these genes in the *redD* mutant is consistent with the precocious erection of aerial hyphae (Figure 1, Figure 2 and Figure 3) and the earlier and enhanced actinorhodin production in this strain (Figure 4).

### 2.3. Survival of S. coelicolor to Prodiginine Production

If PdGs are potent killing factors that induce cell death before the onset of morphogenesis, how does *S*. *coelicolor* survive their production and manage to reach the later stages of its developmental program? The question is very intriguing because the *red* biosynthetic gene cluster (BGC) responsible for PdGs production does not contain genes classically found in BGCs involved in the production of all other DNA-damaging antibiotics [27]. The current hypothesis for resistance to PdGs would involve a stochastic mechanism generating persister filaments that would survive for being in a metabolically dormant state and therefore insensitive to cell death signals triggering biosynthesis of PdGs (see Discussion).

We sought for such viable filaments when *S*. *coelicolor* reached its peak of production of PdGs and cell death after 50 h of incubation. Spores of *S. coelicolor* M145 were spread on the surface of R2YE agar plates, and 0.5-mm thick slices of confluent solid cultures were collected and stained with Syto9 (that stains the DNA of living and dying filaments) as described previously [13]. Confocal fluorescence microscopy allows us to visualize PdG producing filaments by their red autofluorescence (RAF) while Syto9 staining will expose PdG non-producing filaments. In situ visualization of PdGs production revealed important RAF associated with PdG production as previously described [13] (Figure 6a).

Within the death zone on the upper part of a *S*. *coelicolor* culture, Syto9 staining reveals the occurrence of short filaments that do not display any RAF associated with PdG production (Figure 6a). The small proportion of filaments not producing PdGs is difficult to quantify because they are trapped into the dense network of filaments saturated with PdGs. We ignore why these filaments do not produce PdGs but this absence of production is most likely responsible for their survival during the important cell death round that precedes the erection of the aerial structures devoted to the species preservation [50].

At later time points, the absence of PdG biosynthesis in surviving filaments allows the visualization of their emergence from the mass of dying/dead vegetative hyphae (Figure 6b,c) at the beginning of the differentiation process (raising of aerial mycelium). Filaments emanating from the lysing vegetative mycelium do not display RAF suggesting that hyphae surviving the death round preceding sporulation are those that did not produce PdGs. The survival of *S*. *coelicolor* after PdG production would therefore be the consequence of the failure to synthetize PdGs by very small proportion of filaments.

## 3. Discussion

We previously demonstrated that the production of PdG antibiotics plays a role in the PCD of Streptomyces coelicolor [13]. Indeed, when *S. coelicolor* is unable to produce PdGs on the rich R2YE medium, the second and most important round of cell death does not occur [13]. The function of PCD would be to provide nutrients to the developing aerial hyphae, which later give rise to spores. In this report, we examined the impact of the PdG-associated round of PCD on the timing of morphological development (i.e., the raising of aerial hyphae). We found that the inactivation of redD, the transcriptional activator of PdGs biosynthesis, accelerated the onset of morphological development in this organism. This precocious morphogenesis was evident at the phenotypic level (i.e., aerial hyphae formation, Figure 1) and at the level of transcription of key developmental genes (Figure 5). When PdG production was reduced by the external supply of phosphorylated sugars it also resulted in the accelerated onset of sporulation (Figure 3). In contrast, the introduction of pELT003 (*redD* on a low-copy-number plasmid) resulted in increased PdG production which delayed and even blocked development (Figure 2). The accelerated development of the *redD* mutant M510 was also previously suggested in a transcriptomic study of the pathway-specific activators actII-ORFIV and *redD* mutants [51]. Though not phenotypically described and illustrated in their transcriptional analysis, Huang and colleagues also reported the overexpression of sporulation-associated genes such as four of the eight ORFs of the *whiE* locus in the *redD* mutant [51]. 

At this stage one could also postulate that the observed precocious sporulation of the PdG non-producing mutant would not be due to the absence of PdGs but might be the consequence of the altered expression of putative *redD*-dependent sporulation genes. Based on the transcriptomic analysis of the *redD* mutant, in addition to genes of the *red* cluster, only six transcription units—called “ecr” for expression coordinated with red—showed their expression downregulated due to *redD* deletion. Among these were *ecrA1/A2* which disruption revealed to be two-component system involved in the activation of PdG synthesis [52]. The same conclusion was deduced for another two-component system *ecrE1/E2* [53]. The other ORFs, whose expression depends on RedD, are *ecrB* (SCC121.22c), which encodes a putative membrane protein, *ecrC* (SCD12A.15c) and *ecrD* (SCC61A.37c), which, respectively, are proposed to encode an integral membrane ATPase and a secreted protein of unknown function, and finally *ecrF* (SC1A6.12c), which is a hypothetical protein. The biological function of these four additional RedD target genes being unknown, we therefore indeed cannot exclude that one, or some of them could be involved in the morphological differentiation of *S*. *coelicolor*. As an important validation of our hypothesis that RedD acts via PdGs themselves, the chemical inhibition of PdG production by adding phosphorylated sugars to the R2YE medium—and therefore without affecting the *redD* locus—also resulted in accelerated sporulation (Figure 3). This supports the evidence that PdGs are the causing agents that delay the onset of aerial mycelium formation.

How might the absence of a toxic compound accelerate the timing of sporulation? One possibility is that when *S. coelicolor* is impaired in producing PdGs, the abundant still-viable biomass rapidly consumes the remaining nutrients and this starvation initiates the developmental program sooner. Indeed, sporulation represents the culmination of the life(cycle) for sporulating microorganisms. As the onset of this state of dormancy is costly and irreversible, microorganisms must consider all alternatives to survive under hostile conditions before triggering sporulation. One of these alternatives would be the killing of the large majority of the vegetative mycelium in order to lower the consumption of the remaining nutrients and provide nutrients from the dying hyphae to feed the survivors. A similar scenario has been previously documented and connected with bacterial developmental processes, such as the cannibalism in *Bacillus subtilis*, and fratricide in *Streptococcus pneumonia* [32]. In each case the induced cell death of a portion of the population is considered to provide an advantage to the survivors. Not making PdGs makes *S*. *coelicolor* sporulate around 2 days earlier. If sporulation is the end of the life (cycle), it means that under the conditions tested PdGs allow almost to double the lifetime of *S. coelicolor*.

Is the delayed sporulation imposed by PdGs a unique phenomenon or do other DNA-damaging molecules or any other non-secreted antibiotics also interfere with the timing of the onset of the developmental program in other streptomycetes? Although many null mutants for the production of one or more antibiotics have been generated in various *Streptomyces* species, researchers usually did not extend the analysis beyond the production of the studied natural product. However, morphological and chemical differentiation being tightly connected in time and space in streptomycetes, we expect that such mutations would rather frequently affect the onset of spore formation, at least under certain culture conditions. This hypothesis finds evidential support in a recent study of Justin Nodwell’s group who revealed that several compounds targeting the integrity of DNA conferred a sporulation defect (white phenotype of *Streptomyces venezuelae*) at sub-inhibitory concentrations [54].

Importantly, we noticed that a small proportion of the vegetative filaments did not produce PdGs (Figure 6a) and therefore remains viable when the large majority of the mycelium is facing the second and most important round of cell death. Why and how is that part of the mycelium protected from mass destruction? The why appears logical: exclusion from PdG-mediated PCD is likely essential as it ensures the survival of those filaments that eventually become the reproductive aerial hyphae, thus ensuring sporulation. We hypothesize that to avoid self-destruction, these PdG non-producing vegetative filaments enter a metabolic dormant state when signals for triggering PdG synthesis are delivered by the first round of cell death. This dormant state would prevent the activation of PdG production in these compartments, and thus save them from PdG-induced DNA and membrane destruction. How then do these compartments sense the impending cell death? The dormant state may be activated by PCD-released organic and inorganic phosphates that are released into the extracellular medium by lysing filaments that already produced PdGs. Indeed, phosphates inhibit PdG production (Figure 3) and cell death [50]. It is attractive to refer to these PdG non-producing filaments as persisters. In both scenarios described above (passive and active mode of survival), resistance would be observed in a small proportion of isogenic cells but with different phenotypes and responses, which fits with the definition of persisters.

Finally, we never observed the red autofluorescence that was associated with PdG production in aerial hyphae emerging from the vegetative mycelium soaked with PdGs. This result confirms earlier results that the *redD* promoter (which was fused to the gene of the enhanced green fluorescent protein, eGFP) is only active in “aging” vegetative hyphae and not in spore chains, nor in the hyphal compartments adjacent to them [55]. Authors also mentioned that the fluorescence of the “EGFP could only be observed when a proportion of the hyphae had lysed”, further supporting that PdG production correlates in time and space with the cell death of *S*. *coelicolor*. Similarly, the time-space monitoring of RedD fused to enhanced yellow fluorescence protein (eYFP), confirmed that RedD is never expressed in aerial hyphae [56]. Taken together, these data suggest that PdG production is required for properly timing the onset of sporulation, increased or reduced production levels resulting in delayed/blocked or accelerated/precocious sporulation, respectively.

## 4. Materials and Methods

### 4.1. Strains and Culture Conditions

*S. coelicolor* M145 and its *redD* deletion mutant M510 were used as the parental (wild-type) strain and as PdGs non-producer, respectively [57]. Strain ET003 harbouring pELT003 [13] was used as PdGs overproducer. pELT003 is plJ2587 harboring *redD* (SCO5877) [58], with its native upstream region (567 bp). Phenotypic studies were performed on R2YE medium agar plates [59]. Plates were inoculated with 500 µL of a 2.10^7^ cfu/mL spore suspension.

### 4.2. Antibiotic Production

Production of PdG was measured as described previously [60]. PdGs were extracted from 4 mg of mycelium of the different *S*. *coelicolor* strains (either from isolated colonies or confluent lawns, see Figure 2) with 2 × 200 µL of methanol (pH 2.0). Absorbance measurements were made at 530 nm, and the semi-quantitative PdG production was calculated according to the level produced by the parental strain *S*. *coelicolor* M145 (fixed to 100%). Production of extracellular actinorhodin-related blue pigments was performed as described in [61]. Extraction of actinorhodin from the agar spent was performed with 1 volume of methanol:chloroform (1:1) then acidified with 1 N HCl to pH 2.0. The chloroform phase was collected, and the absorbance at 650 nm was measured. The mycelium was scraped from the surface (covered with cellophane discs) of the agar plate and the biomass was quantified for normalization of actinorhodin production.

### 4.3. Transcriptional Analysis

Transcription of genes of *S. coelicolor* M145 and its *redD* null mutant M510 was analyzed as described previously [19,62]. Whole colonies grown for 68 h on R2YE agar plates were collected with a razor blade and immediately ground in liquid nitrogen using a mortar and pestle. The quantification of mRNAs in cell lysates was performed via the Nanostring nCounter analysis system. Transcript counts for three genes (*hrdB*, *folB*, and *def*) were used for geometric mean normalization to correct for differences in total mRNA concentration All data were collected from three biological replicates and gene expression was considered significantly altered if mutant counts had a *p* value ≤ 0.02 in a student’s T-test when compared with the wild-type.

### 4.4. Scanning Electron Microscopy

The morphological study of surface-grown vegetative and aerial hyphae of *S. coelicolor* M145 (wild-type) and M510 (∆*redD*) by cryo-scanning electron microscopy (JEOL JSM-840A) was performed as previously described [63].

### 4.5. In Situ Visualization by Confocal Fluorescence Microscopy

Samples were prepared as described previously [13,62]. Observations were made with a Leica TCS-SP2 confocal laser-scanning microscope (Leica Microsystems, Heidelberg, Germany). Samples were excited at a wavelength of 543 nm and emission was examined between 560 and 620 nm as described previously. Image processing and 3D reconstruction of *Streptomyces* cells producing PdGs were performed as described previously. Z-Stacks of confocal images (objective HCX PL APO 63 × 1.20 W CORR UV and pixels 512 × 512) were processed using the Fiji (ImageJ) software. For analyses of dense culture across a transversal section, ten Z-stack images from a 238.1 μm section were used for a standard deviation Z-projection (for light images) or a maximal Z-projection after applying a Gaussian Blur filter with a radius of 1 (for RAF images). For the 3D confocal image stack reconstitution, UCSF Chimera software was used to visualize 84 z-stack images from a 85.2 μm section (objective HCX PL APO 63 × 1.20 W CORR UV with a zoom of 2.8 and pixels 512 × 512) preprocessed by the Fiji software (Gaussian Blur filter for RAF image stack).

## 5. Conclusions

Life for streptomycetes is a cycle that starts with the germination of a spore and ends by the formation of chains of spores. Their development includes a multiphasic growth and complex differentiations where the onset of the sporulation process signifies entering a dormant state until spores encounter more favorable conditions. The red-pigmented tri-pyrroles prodiginines participate to the most important death round of *S*. *coelicolor* and we investigated here how the inhibition or instead the overproduction of these cytotoxic molecules influenced the timing of sporulation of the producer. Inhibition of PdG production resulted in accelerated development whereas their overproduction led to a developmental arrest at the vegetative mycelium state. These results confirm that non-secreted anti-proliferative cytotoxic compounds influence importantly the physiology and development of the producing organism. Our work suggest that microorganisms could use compounds with cytotoxic antiproliferative activity as we use them in cancer therapy: to induce cell death in order to prolongate the lifetime, which means to postpone the onset of morphological differentiation for sporulating bacteria.

## Figures and Tables

**Figure 1 antibiotics-09-00847-f001:**
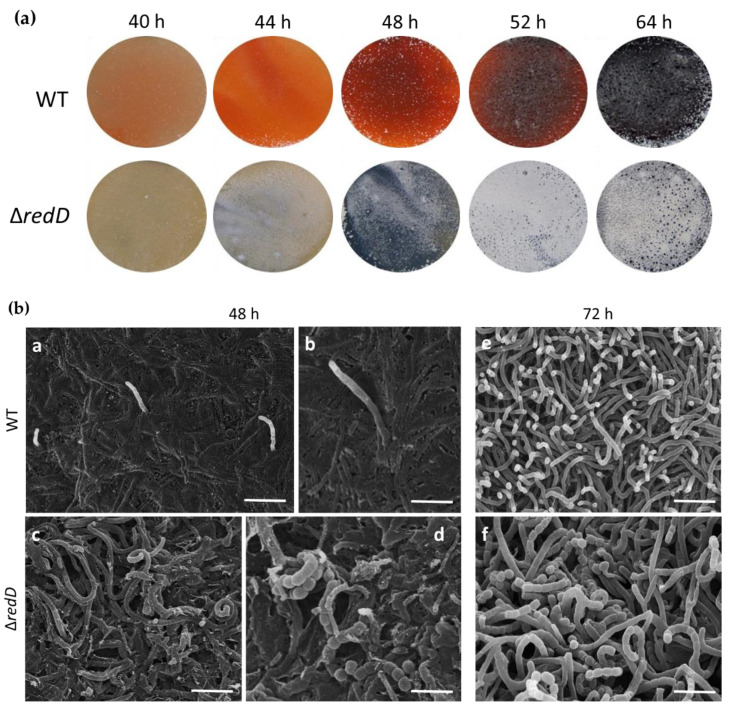
Effect of the *redD* inactivation on morphological differentiation of *S. coelicolor*. (**a**) Pictures of R2YE plates inoculated with *S*. *coelicolor* M145 (WT, wild-type) and its *redD* mutant M510. Note the precocious appearance of white-pigmented aerial hyphae. (**b**) Scanning electron micrographs showing precocious sporulation of the *S. coelicolor redD* null mutant. In (**a**,**b**) (48 h) we can see in the WT the very first aerial hyphae emerging from the dense network of vegetative filaments while in strain M510 (∆*redD*; **c**,**d**) many aerial hyphae have already been produced with occasionally fully matured spore chains (**d**). In (**e**,**f**) (72 h) the *redD* mutant has fully differentiated into mature spore chains whereas the WT only displays erect aerial hyphae. Bars = 10 µm (**a**,**c**); = 3 µm (**b**,**d**); = 5 µm (**e**,**f**).

**Figure 2 antibiotics-09-00847-f002:**
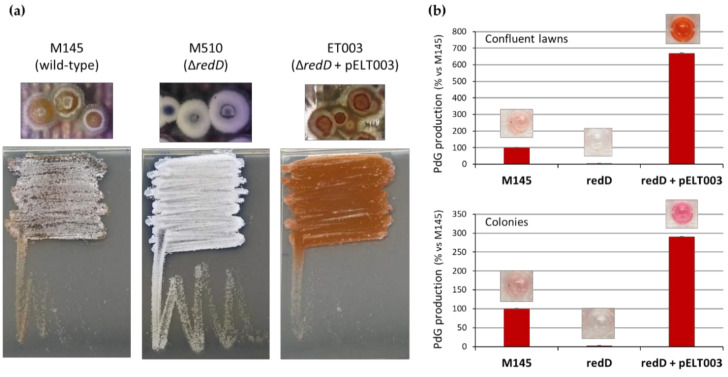
Multiple copies of *redD* results in higher PdG production, thereby blocking sporulation of *S*. *coelicolor*. (**a**) Phenotypes of 65 h old confluent lawns and single colonies of *S*. *coelicolor* M145 (wild-type, parental strain), M510 (*redD* deletion mutant), and strain ET003 (*redD* on the low-copy-number plasmid pIJ2587). (**b**) Semi-quantitative measurement of PdG synthesis in the three studied *S*. *coelicolor* strains revealing the overproduction of PdGs in ET003. Note that thiostreptone was not included in the medium for optimal comparison of strain ET003 with strains M145 and M510 that do not possess the *tsr* resistant gene present in pELT003.

**Figure 3 antibiotics-09-00847-f003:**
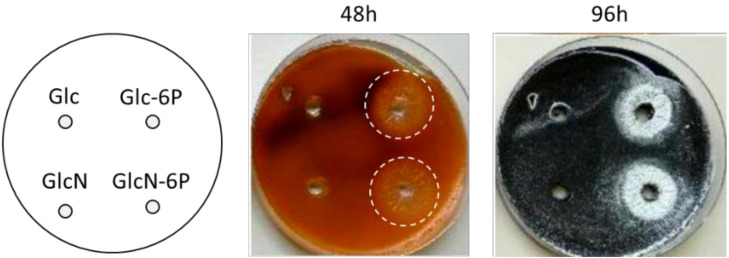
Sugar phosphates reduce PdG production and accelerate sporulation of *S. coelicolor*. A suspension containing 10^7^ viable spores of *S. coelicolor* M145 was plated on R2YE solid medium. After 20 h of growth, wells (5 mm diameter) were made into the agar and 100 mM of glucosamine (GlcN), glucose (Glc), glucosamine-6-phosphate (GlcN-6P), and glucose-6-phosphate (Glc-6P) were deposited into them. Dotted circles highlight the zone around the source of phosphorylated sugars with reduced PdG production. Note how zones with reduced production of PdGs at 48 h coincides with the zones with precocious erection of aerial hyphae at 96 h.

**Figure 4 antibiotics-09-00847-f004:**
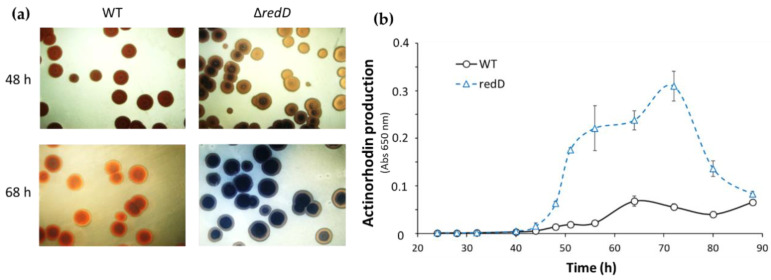
Comparison of actinorhodin production between *S. coelicolor* M145 (WT) and the ∆*redD* mutant. (**a**) Production of the blue-pigmented actinorhodin on isolated colonies. Pictures show colonies from the reverse side of the plates. (**b**) Semi-quantitative assessment of actinorhodin production between *S. coelicolor* M145 (WT) and the ∆*redD* mutant grown in R2YE agar plates.

**Figure 5 antibiotics-09-00847-f005:**
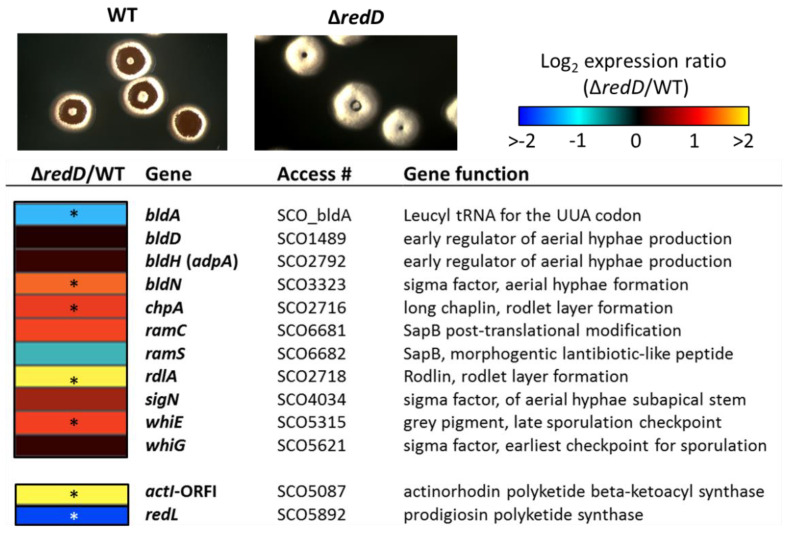
Transcription of developmental genes in *S. coelicolor* wild-type (M145) and its ∆*redD* mutant (M510). Gene names are shown at right of heat map; color legend for Log_2_ expression (fold change) is shown above. Genes were considered significantly regulated if they had *p* values ≤ 0.02 in Student’s T-test (three replicates) when compared between wild-type and mutant strains. Genes meeting this criterion are noted with an asterisk (*). Colonies of *S. coelicolor* M145 (WT) and M510 (Δ*redD*) were around 5 mm in diameter.

**Figure 6 antibiotics-09-00847-f006:**
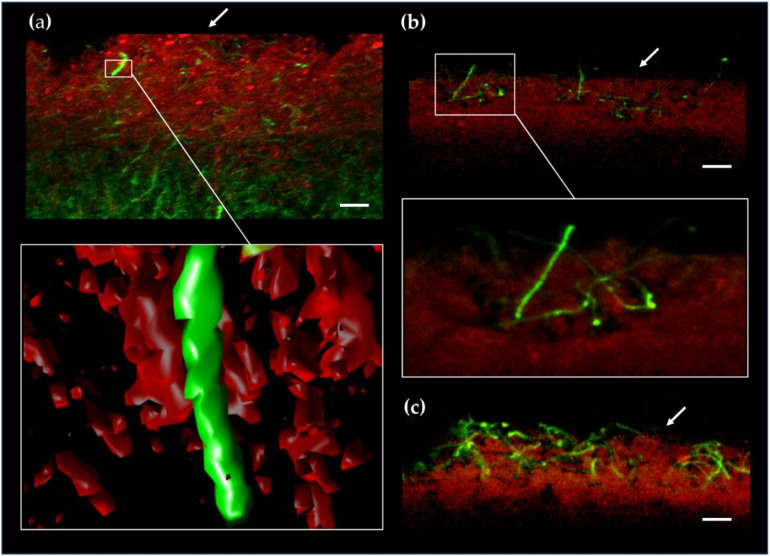
Aerial growth of PdG non-producing filaments in *S*. *coelicolor*. Fluorescence confocal micrographs (3D reconstruction) of a transverse section of a *S*. *coelicolor* culture grown for 40 h on R2YE agar plates. Red autofluorescence (RAF) associated with PdG production (RAF, red-colored), and Syto9 staining of PdG non-producing filaments (green-colored, SYTO9). Note that Syto9 stains the DNA of both living and dying filaments unless PdG production is too abundant, which leads to DNA destruction thereby preventing intercalation of dyes). (**a**) Note the very rare occurrence of Syto9 stained filaments in the zone that massively produces PdGs. Close up and 3D reconstruction of a filament that does not produce PdGs. (**b**,**c**) Aerial hyphae emerging from the surface of the vegetative mycelium after 70 h (**b**) and 90 h (**c**) of growth. Filaments emanating from the lysing vegetative mycelium do not display RAF, suggesting that hyphae surviving the death round preceding morphological differentiation are those that did not produce PdGs.

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
