# Peer review of "Prodiginines Postpone the Onset of Sporulation in Streptomyces coelicolor"

_antibiotics, 2020, doi:10.3390/antibiotics9120847_

Round 1
Reviewer 1 Report
The authors here look at the role of prodiginines in what they are calling “programmed cell death” (which may be a misnomer). They present some interesting evidence linking prodiginine production with the developmental pathways in S. coelicolor A3(2) in figures 1-3, but it is not clear from the manuscript as written if 1)the authors have performed the necessary controls (see point 3 below), or 2)whether this link is causal, or merely a correlation. At a bare minimum, it is necessary to complement a mutant strain such as the redD mutant strain used here. Overall, the authors should be more cautious in their interpretation of their results.
- Use of the term “programmed cell death” implies that there is in fact, a program directing cell death during sporulation in Streptomyces: if such were the case, one would expect to isolate a number of mutant strains deficient in this process. (Perhaps the authors would argue that the redD mutant strain used here is such a mutant, but they don’t present any evidence supporting this; and it is not clear, from evidence presented here, whether redD is directly involved in the pathway or merely correlated temporally with it.) I would encourage the authors to be more rigorous in their use of terms such as PCD and persisters (see below).
- As the authors themselves point out, Red is not ubiquitous amongst Streptomyces: any general pathway for so-called “programmed cell death” is likely to involve more conserved genes. It is not clear what relevance this study has beyond Streptomyces coelicolor A3(2).
- The redD mutant has not been complemented (Fig 1, 3, 4) – this is a necessary control to show that the phenotypes shown are due to deletion of redD itself and not due to some other mutation introduced during strain construction.
- As redD is a regulatory gene located within the Red cluster, the authors seem to implicitly assume that the only role of RedD is in regulation of Red biosynthesis. To the best of my knowledge, this has not yet been formally proven: it is possible that RedD has additional regulatory targets outside prodiginine biosynthesis. Based on the data presented, the authors cannot rule out the possibility that another RedD target gene is involved in the phenotypes observed here. It would simplify the analysis and interpretation of their results considerably if the authors instead used a mutant strain lacking a structural gene essential for prodiginine biosynthesis.
- The authors do not present any evidence that prodiginine non-production is the reason for survival of the “persister” filaments (Fig. 5) – this could simply be correlation, not cause-and-effect. Do these cells die because they are producing prodiginine, or do these stressed, dying cells produce prodiginine because they are dying?
Equally, this correlation between prodiginine biosynthesis and cell death could be an indirect effect - presumably prodiginine biosynthesis has substantial effects on e.g. metabolism and carbon flux throughout the population.
Moreover, the authors should avoid the use of the term “persister” as it is confusing with respect to actual persister cells (which these have not been proven to be: indeed, Fig 5 suggests that they may be actively growing)
Minor points:
- Figure 1 legend does not specify which medium the cells were grown on for SEM (R2YE?)
- Figure 2 lacks a solvent control (e.g. a well containing water or the solvent used to dissolve the phosphorylated sugars)
- Fig 3 – not clear how/if cells were normalized for measuring Act production (cell dry weight?) Normalization essential to ensure that you are not simply using more cells from one strain than another and thus seeing higher amounts of Act.
- Fig 3b – not clear from the text/methods if Act production was measured in cells grown in liquid or on solid medium
- Figure 4/methods – not clear how/if cells were normalized (total RNA? cell mass?) – gene expression differences could be due to differing amounts of input
- Figure 5 – authors must avoid using red/green as it is inaccessible to color-blind readers
- Earlier studies have shown that RedD expression is confined to aging vegetative mycelia (using a fluorescent reporter) – it seems strange that the authors do not cite this work in their discussion of Figure 5 (Sun et al 1999).
- Minor editing for grammar and spelling needed throughout
Reviewer 2 Report
This is an interesting manuscript. The Introduction is fine, the results are well presented and the discussion is relevant. A more voluminous Materials and Methods section would be preferable, so that the reader should not have to go to the literature for information. There are some English grammar mistakes.
Minor points
Line 21: Abbreviations must be defined the first time they are mentioned
L 25: bldN: Abbreviations must be defined the first time they are mentioned
L 36+37: The most prolific microbial source of these natural products are; change to: The most prolific microbial source of these natural products is
L54: …environments that is also home; change to: …environments that are also
L65: Amongst these type of compounds; change to: Amongst this type of compounds
L104:(WT) must be defined
L109: maturated; change to: matured
L116: (Figure 2b, panel f); change to: (Figure 1b, panel f)
L137: Pictures shows; change to: Pictures show
L169: the lantibiotic-like peptide; change to: the antibiotic-like peptide
L193: how does S.coelicolor survives; change to: how does S.coelicolor survive
L431: Ref 45? Something is missing in this reference?
Reviewer 3 Report
The manuscript entitled “Prodiginines postpone the onset of sporulation in Streptomyces coelicolor” is within the scope of Antibiotics. The paper is overall well-written and the experimental design is well formulated. I have some suggestions, which are the following:
Abstract: for clarity, the abstract should be structured (see instructions for Authors).
Figure 1: Figure 1 is unclear. Figures a and b should be separated with their respective captions.
Discussion: the discussion is too short. It must be broadned and argued.
References: the manuscript does not include all relevant references. Several interesting works regarding prodiginines are not cited in the manuscript:
Stanley, A. E., Walton, L. J., Zerikly, M. K., Corre, C., & Challis, G. L. (2006). Elucidation of the Streptomyces coelicolor pathway to 4-methoxy-2, 2′-bipyrrole-5-carboxaldehyde, an intermediate in prodiginine biosynthesis. Chemical communications, (38), 3981-3983.
Singh, R., Mo, S., Florova, G., & Reynolds, K. A. (2012). Streptomyces coelicolor RedP and FabH enzymes, initiating undecylprodiginine and fatty acid biosynthesis, exhibit distinct acyl-CoA and malonyl-acyl carrier protein substrate specificities. FEMS microbiology letters, 328(1), 32-38.
Cerdeño, A. M., Bibb, M. J., & Challis, G. L. (2001). Analysis of the prodiginine biosynthesis gene cluster of Streptomyces coelicolor A3 (2): new mechanisms for chain initiation and termination in modular multienzymes. Chemistry & biology, 8(8), 817-829.
Lim, Y., Jung, E. S., Lee, J. H., Kim, E. J., Hong, S. J., Lee, Y. H., & Lee, C. H. (2018). Non-targeted metabolomics unravels a media-dependent prodiginines production pathway in Streptomyces coelicolor A3 (2). PloS one, 13(11), e0207541.
Pérez-Tomás, R., & Vinas, M. (2010). New insights on the antitumoral properties of prodiginines. Current medicinal chemistry, 17(21), 2222-2231.
Williamson, N. R., Fineran, P. C., Leeper, F. J., & Salmond, G. P. (2006). The biosynthesis and regulation of bacterial prodiginines. Nature Reviews Microbiology, 4(12), 887-899.
Reviewer 4 Report
This manuscript seems to present results already published by the authors with some additional information.
The experiments shown should also be presenting data from the complemented redD mutant strain, in order to ensure that the results observed are due to the absense of RedD and not due to additional mutations on the redD mutant strain.
